# Insight into the Structural, Electronic, Elastic, Optical, and Magnetic Properties of Cubic Fluoroperovskites *ABF*_3_ (A = Tl, B = Nb, V) Compounds: Probed by DFT

**DOI:** 10.3390/ma15165684

**Published:** 2022-08-18

**Authors:** Saima Ahmad Shah, Mudasser Husain, Nasir Rahman, Mohammad Sohail, Rajwali Khan, Abed Alataway, Ahmed Z. Dewidar, Hosam O. Elansary, Lamia Abu El Maati, Kowiyou Yessoufou, Asad Ullah, Aurangzeb Khan

**Affiliations:** 1Department of Physics, Abdul Wali Khan University, Mardan 23200, Pakistan; 2Department of Physics, Shaheed Benazir Bhutto Women University, Peshawar 00384, Pakistan; 3Department of Physics, University of Lakki Marwat, Lakki Marwat 28420, Pakistan; 4Prince Sultan Bin Abdulaziz International Prize for Water Chair, Prince Sultan Institute for Environmental, Water and Desert Research, King Saud University, Riyadh 11451, Saudi Arabia; 5Department of Agricultural Engineering, College of Food and Agriculture Sciences, King Saud University, Riyadh 11451, Saudi Arabia; 6Plant Production Department, College of Food & Agriculture Sciences, King Saud University, Riyadh 11451, Saudi Arabia; 7Floriculture, Ornamental Horticulture, and Garden Design Department, Faculty of Agriculture (El-Shatby), Alexandria University, Alexandria 21545, Egypt; 8Department of Geography, Environmental Management, and Energy Studies, University of Johannesburg, APK Campus, Johannesburg 2006, South Africa; 9Department of Physics, College of Science, Princess Nourah bint Abdulrahman University, P.O. Box 84428, Riyadh 11681, Saudi Arabia; 10Department of Mathematical Sciences, University of Lakki Marwat, Lakki Marwat 28420, Pakistan; 11University of Lakki Marwat, Lakki Marwat 28420, Pakistan

**Keywords:** DFT, fluoroperovskites, elastic properties, structural properties, electronic properties

## Abstract

This work displays the structural, electronic, elastic, optical, and magnetic properties in spin-polarized configurations for cubic fluoroperovskite $ABF_{3}$ (A = Tl, B = Nb, V) compounds studied by density functional theory (DFT) by means of the Tran-Blaha-modified Becke-Johnson (TB-mBJ) approach. The ground state characteristics of these compounds, i.e., the lattice parameters $a_{0}$, bulk modulus (B), and its pressure derivative $\left(B\prime \right)$ are investigated. The structural properties depict that the selected compounds retain a cubic crystalline structure and have stable ground state energy. Electronic-band structures and DOS (density of states) in spin-polarized cases are studied which reports the semiconducting nature of both materials. The TDOS (total density of states) and PDOS (partial density of states) studies in both spin configurations show that the maximum contributions of states to the different bands is due to the B-site (p-states) atoms as well as F (p-states) atoms. Elastic properties including anisotropy factor (A), elastic constants, i.e., *C*_11_, *C*_12_, and *C*_44_, Poisson’s ratio (υ), shear modulus and (G), Young’s modulus (E) are computed. In terms of elastic properties, the higher (bulk modulus) “B” and ratio of “B/G” yield that these materials exhibit a ductile character. Magnetic properties indicate that both the compounds are ferromagnetic. In addition, investigations of the optical spectra including the real ($\varepsilon_{1}\left(\omega \right)$) and imaginary ($\varepsilon_{2}\left(\omega \right)$) component of the dielectric function, refractive index $n\left(\omega \right)$, optical reflectivity $R\left(\omega \right)$, optical conductivity $\sigma \left(\omega \right)$, absorption coefficient $\alpha \left(\omega \right)$, energy loss function $L\left(\omega \right)$, and electron extinction coefficient $k\left(\omega \right)$ are carried out which shows the transparent nature of $TlVF_{3}$ and $TlNbF_{3}$. Based on the reported research work on these selected materials, their applications can be predicted in many modern electronic gadgets.

## 1. Introduction

Recently the most prevalent and extensively researched structure in materials science is the perovskite structure. Fluoroperovskites *ABF*_3_ compounds are interesting nowadays for researchers due to their wide applications in many semiconducting industries, solar cell industries, and other electronic gadgets. The fluoroperovskites with A and B are metallic cations and F, which is a highly electronegative anion, possess various structural, electronic, elastic, thermoelectric, thermodynamic, and magnetic properties because of its variant electronic band gaps. The generalized perovskites compounds have the chemical formula ABX_3_, in which A, B are two cations of various magnitudes, and X is an anion linked to both of them. Its ideal structure is cubic, and the B atoms in a typical anionic octahedron are located in the middle. This atomic configuration may appear simple, but it conceals a variety of unique physical and chemical features. The selected fluoroperovskite $ABF_{3}$ (A = Tl, B = Nb, V) compounds possess a cubic symmetry. In the $pm-3m\;\left(\#221\right)$ space group, “A” and “B” atoms (cations) occupy the edges and body center positions respectively while the face centers are occupied by “F” atoms (anion) [1,2,3]. It is reported that most of the cubic fluoroperovskite compounds are elastically anisotropic and mechanically stable and also possess interesting electronic properties and magnetic properties [4]. Due to their attractive properties, these materials have a variety of applications, including photovoltaic, vehicle energy, device, and the lenses industry as well as their transparent characteristics, which are used in antireflection coatings [5,6,7,8,9]. Woodward and Lufaso reported that cubic perovskite can transfer to other forms of crystal structures [10]. The combination of F with either inorganic or organic and transition metals form stable fluoroperovskites [11]. In this work, the elastic, structural, electronic, and optical properties within the spin-polarized case of $TlVF_{3}$ and $TlNbF_{3}$ compounds have been studied theoretically by DFT (TB-mBJ method) using the WIEN2K computational simulation code. The method of TB-mBJ potential is used because of its accuracy in the electronic band gaps, as the LDA or GGA exchange-correlation functional underestimates the electronic band gaps. These compounds have not been studied theoretically or experimentally before. Therefore, this work can be used as a reference for further studies of such types of compounds.

## 2. Computational Methodology

In the present work, the calculations are done with TB-mBJ potentials as coupled with GGA [12] installed on the DFT and applied in the WIEN2K code [13]. The structural, elastic, electronic, and elastic properties are investigated within the spin-polarized configuration. The k-points used are chosen as 2000. The structural basic parameters are computed from the Birch–Murnaghan equation of state that optimized the energy versus volume. The computation of elastic properties is done using the IRelast package, which is an interface within the WIEN2K. For the plotting purpose of the present study, the Xmgrace, Xcrysden, origin is chiefly used. The core and valence energy gap 0f −6 Ry is selected to avoid the charge leakage from the atomic spheres. Furthermore, the value of RMT × K_max_ is chosen to be 7, which is a suitable criterion for convergence.

## 3. Results and Discussion

### 3.1. Structural Properties

The unit cells for both the cubic fluoroperovskites $TlVF_{3}$ and $TlNbF_{3}$ having space group Pm-3m (#221) are displayed in Figure 1. In $TlVF_{3}$ and $TlNbF_{3}$ unit cells, the occupied Wyckoff positions are Tl = (0 0 0), V or Nb = (0.5, 0.5, 0.5) and F at (0, 0.5, 0.5), (0.5, 0, 0.5), and (0, 0, and 0.5).

The Figure 2 shows the optimized energy curves for both the cubic $TlVF_{3}$ and $TlNbF_{3}$ fluoroperovskite structures using the TB-mBJ method by fitting the Birch–Murnaghan equations of state. These calculations predict the ground-state energy, ground-state volume ($V_{0}$), and B and B′ of the structure. The ground state of the system can be determined by observing the points in the Birch–Murnaghan fitted curve possessing the lowest energy relative to the volume, and the parameters, i.e., energy and volume related to the point are regarded as the ground-state energy and ground-state volume [14,15,16,17]. Lattice constants can be determined by using the ground-state volume. The obtained results are summarized in Table 1.

### 3.2. Elastic Properties

The Hyperplastic Materials possess 21 independent elastic constants (*C_ij_*) with stress-strain symmetry [18]. However, for cubic crystals, this symmetry decreases to only three elastic moduli, *C*_11_, *C*_12,_ and *C*_44_ [19,20]. Since these elastic constants of solids give information regarding the response of crystals to external forces, especially the mechanical strength of the material. Once the crystal’s elastic constants *C*_11_, *C*_12_, and *C*_44_ are determined, the other parameters, i.e., B, G, υ, A, E, and B/G of the solid crystals can be calculated by making use of Voigt–Reuss–Hill Equations (1)–(6) [21,22]. The calculated results are summarized as:(1)$$A=\frac{2C_{44}}{C_{11-C_{12}}}$$
(2)$$E=\frac{9GB}{3B+G}$$
(3)$$\upsilon =\frac{3B-2G}{2\left(2B+G\right)}$$
(4)$$G=\frac{1}{2}\left(G_{\upsilon }+G_{R}\right)$$
(5)$$G_{v}=\frac{1}{5}\left(C_{11}-C_{12}+3C_{44}\right)$$
(6)$$G_{R}=\frac{5C_{44}\left(C_{11}-C_{12}\right)}{4C_{44}+3\left(C_{11}-C_{12}\right)}$$

In the above equations “A” is the anisotropy factor, “E” shows Young’s modulus, “υ” is the Poisson ratio, “G” depicts the shear modulus, “G_v_” is Voigt’s shear modulus, and the “G_R_” represents Reuss’s shear modulus.

The high value of “B” reflects a tendency for better ductility (brittleness). The obtained values of B for $TlNbF_{3}$ is higher than $TlVF_{3}$. This shows that the compressibility of $TlNbF_{3}$ is less than $TlVF_{3}$. The resistance to the change in the shape of the solids is determined by the shear modulus G [23]. The calculated values of the cubic fluoroperovskites, $TlVF_{3}$ and $TlNbF_{3}$ are 5.8491 and −15.332 GPa, respectively. This shows that the shear modulus for $TlVF_{3}$ is greater than for $TlNbF_{3}$. The higher shear moduli values are a better predictor of the hardness of materials. Anisotropy factor “A” is the rate of the degree of elastic anisotropy in a cubic crystal. The material will be completely isotropic if A = 1. However for A > 1 or A < 1 the crystal will be anisotropic [24]. It can be observed in Table 2 that the obtained values of the anisotropy factors are −0.0744 and −0.3804 for $TlVF_{3}$ and $TlNbF_{3}$, respectively, which suggests the anisotropic behavior of the crystals. The bonding forces are determined by Poisson’s ratio. This ratio is <0.1 for materials possessing covalent bonding, while for υ = 0.25, the ionic character will be present [25]. The obtained Poisson’s ratio values are 0.6725 and 0.8816 for $TlVF_{3}$ and $TlNbF_{3}$, respectively. This indicates the ionic character of $TlVF_{3}$ and $TlNbF_{3}$. The “E” defines the stiffness of the solid material. These calculated values of Young’s modulus [26] are presented for both the $TlVF_{3}$ and $TlNbF_{3}$.

The materials possess ductility if the ratio of “B/G” is greater than 1.75, if not then it must be brittle. As can be seen in Table 2, the obtained results are 10.8441 and 7.1454 for $TlVF_{3}$ and $TlVF_{3}$ respectively. Hence, we can deduce that $TlVF_{3}$ and $TlNbF_{3}$ possess ductile nature.

### 3.3. Electronic Properties

Bands structure alongside high symmetries directions in the first Brillouin zone for $TlVF_{3}$ and $TlNbF_{3}$ are predicted with TB-mBJ in both spin schemes (Spin up and Spin down) as shown in Figure 3 for bands structures of (a) *TlNbF*_3_ (b) *TlVF*_3._

It can be seen that both $TlVF_{3}$ and $TlNbF_{3}$ possess band gaps within both spin configurations and are therefore semiconductors in nature.

To further explain the electronic band structure the TDOS and PDOS are considered in both spin configurations. It is detectable from Figure 4 that the major contributions of the states in the valence band of $\mathrm{the}\;TlVF_{3}$ compound is due to the V (p-states) atom, and the contributions of the F (p-states) atom, while a very small contribution is that of Tl atoms. Similarly, from Figure 4, it can be seen that in compound $TlNbF_{3}$ the largest contribution of states is due to the participation of F (p-states) and Nb (d-states) is also involved. In the latter compound, the situation is opposite to that of the first compound. It is also very obvious from Figure 4 that the maximum contribution to the density of states in conduction bands occurs because of the Tl, V, and Nb atoms.

### 3.4. Magnetic Properties

The DOS is associated with a high exchange splitting of the X atoms’ three d-states, resulting in enormous spin moments at their locations and approximately a total magnetic moment of 4.12 µ_B_, possessed by *TlNbF_3_* and 3.294 µ_B_ for *TlVF_3_* and is recorded in Table 3. It is very obvious from the values of magnetic moments listed in Table 3 that Tl and F have a smaller contribution to the total magnetic moments, and apart from that, the Nb and V sites possess a leading contribution to the total magnetic moments. As the total magnetic moments for both the compounds are greater than one, therefore these compounds are strongly ferromagnetic. The interstitial site is the set of points of space that are not in any of the atomic spheres. Therfore, the spin interstitial magnetic moment is the variance amongst the number of spin-up and spin-down electrons in the interstitial. Such sites also carry sizable local spin magnetic moments in different ranges.

The prediction of the ferromagnetic behavior of these materials is applicable in many memory storage devices and other modern electronic gadgets.

### 3.5. Optical Properties

The dielectric function represents the response of the material at given photon energy given by:(7)$$\varepsilon \left(\omega \right)=\varepsilon_{1}\left(\omega \right)+i\varepsilon_{2}\left(\omega \right)$$

The variation of the real part of the dielectric function $\varepsilon_{1}\left(\omega \right)$, which exhibits the electronic polarizability information of the material, with the incident photon energy up to 13 eV is depicted in Figure 5a. It is clear from Figure 5a that the dielectric constant at zero frequency limit $\varepsilon_{1}\left(0\right)$ of $TlVF_{3}$ and $TlNbF_{3}$ is found to be infinite. Exceeding the zero-frequency limit $\varepsilon_{1}\left(\omega \right)$ decreases sharply and reaches a minimum value of 4.46 at 5.42 eV for *TlNbF_3_* and 4.47 at 5.49 eV. Further increasing the plots increases sharply and reaches a maximum of 3.15 at particular energy of 3.96 eV and 4.18 at 5.96 eV for $TlVF_{3}$ and $TlNbF_{3}$, respectively. For negative values of $\varepsilon_{1}$(ω), both $TlVF_{3}$ and $TlNbF_{3}$ compounds are characterized as semiconductors, and lose their dielectric properties.

Figure 5b shows the imaginary part of the frequency-dependent dielectric function $\varepsilon_{2}\left(\omega \right)$. The threshold energy (also known as absorption edge) is 34.98 eV and 12.69 eV for *TlVF_3_* and *TlNbF_3_*, respectively. The maximum absorption peak of the dielectric function occurred at 42.30 eV for $TlVF_{3}$ and 15.29 eV for $F_{3}$.

The refractive index is the measure of the refraction of light. This parameter is extensively used in photoelectric applications. It consists of real ($n\left(\omega \right)$: refractive index) and imaginary part ($k\left(\omega \right)$: extinction coefficient). Both the calculated parameters for $TlVF_{3}$ and $TlNbF_{3}$ compounds are depicted in Figure 5c and Figure 6. The static refractive index $n\left(0\right)$ are 6.9 and 4.6 for $TlVF_{3}$ and $TlNbF_{3}$, respectively, while the maximum values of $n\left(\omega \right)$ are 1.8 at 4.02 eV for $TlVF_{3}$ and 2.06 at 5.98 eV for $TlNbF_{3}$. Here, as $n\left(\omega \right)>1$, therefore the electrons slow down when they enter a denser medium due to the interaction with electrons. The $n\left(\omega \right)$ will be high if the electrons’ speed becomes smaller when it enters the material medium. It should be noted that the increases in the electron density in the material will also increase the $n\left(\omega \right)$. However, $n\left(\omega \right)$ is related to the bonding characteristics as well. Generally speaking, the refractive index of ionic compounds is lower than that of covalent compounds. In covalent bonds, ions share more electrons than ionic bonds. Therefore, due to the high density of electrons, the large number of photons will interact with the photons (photons) to slow down.

The reflectivity $R\left(\omega \right)$ spectrum of both the compounds is presented in Figure 5d. $R\left(0\right)$ for $TlVF_{3}$ is 60% while for $TlNbF_{3}$ it is 45%. However, the maximum value of $R\left(\omega \right)$ is 14% at 4.1 eV for $TlVF_{3}$ and 45% at 12.6 eV for $TlNbF_{3}$.

The optical conductivity $\sigma \left(\omega \right)$ of $TlVF_{3}$ and $TlNbF_{3}$ can be seen in Figure 6. The maximum value of $\sigma \left(\omega \right)$ for both the compounds $TlVF_{3}$ and $TlNbF_{3}$ is obtained which is 1.7 Ω^−1^cm^−1^ at 9.1 eV and 4.2 Ω^−1^cm^−1^ at 8.9 eV for $TlVF_{3}$ and $TlNbF_{3}$, respectively. Similarly, the results observed for absorption coefficient $\alpha \left(\omega \right)$ are depicted in Figure 6. The maximum of $\alpha \left(\omega \right)$ occurs at 0.34 eV and 4.2 eV for $TlVF_{3}$ and $TlNbF_{3}$, respectively, making the materials attractive for applications in optoelectronic devices.

## 4. Conclusions

To summarize, the structural, electronic, elastic, magnetic, and optical properties of ABF3 (A = Tl, B = Nb, V) compounds were investigated by density functional theory (DFT). The lattice parameters, i.e., $a_{o}$ and the $E_{o}$ decreases, while B and $B^{\prime }$ increase with the replacement of the cation, i.e., V to Nb. Both the compounds are semiconductors and possess ductile and anisotropic properties. The TDOS and PDOS results show that the major contribution to the states in both the compounds is due to the B site atoms, i.e., V and Nb and less from F atoms orbitals. The bonding nature of both the compounds is ionic. Furthermore, the calculated results of elastic properties reveal the mechanical stability of both the compounds $TlVF_{3}$ and $TlNbF_{3}$ which have applications in high-performance electronic devices. Investigation of magnetic properties indicates that these compounds have integral magnetic moments and are thus classified as ferromagnetic. The optical properties results show that $TlVF_{3}$ and $TlNbF_{3}$ are transparent to incident photons, which makes them suitable for lenses and anti-reflection coatings.

## Figures and Tables

**Figure 1 materials-15-05684-f001:**
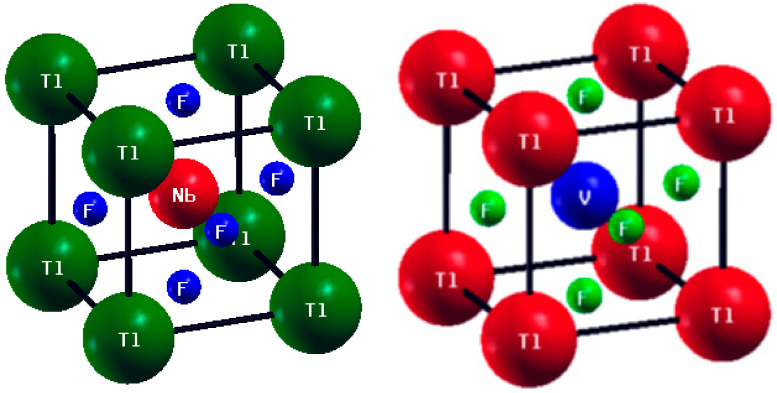
Crystal structures of $ABF_{3}$ (A = Tl, B = Nb, V) compounds.

**Figure 2 materials-15-05684-f002:**
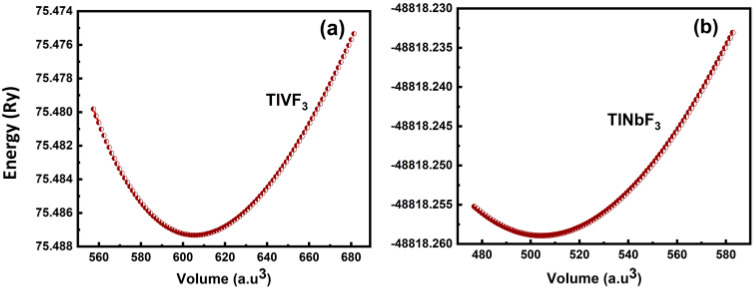
Total energy vs. volume of cubic (**a**) *TlVF*_3_ (**b**) *TlNbF*_3_.

**Figure 3 materials-15-05684-f003:**
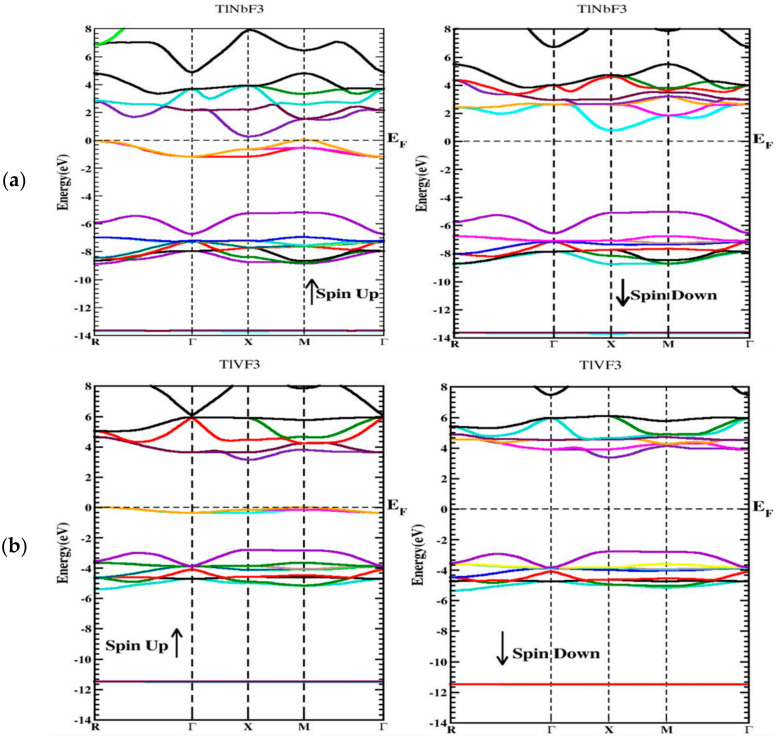
Band configurations of (**a**) *TlNbF*_3_ (**b**) *TlVF*_3_.

**Figure 4 materials-15-05684-f004:**
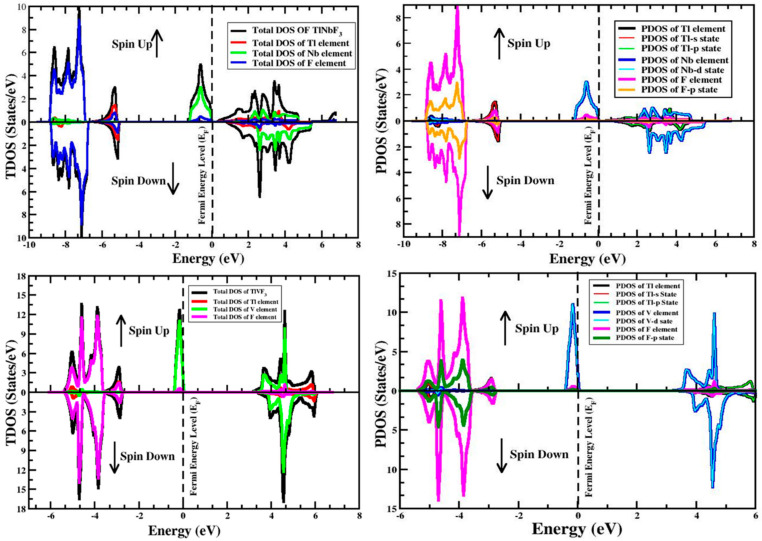
TDOS and PDOS of *TlVF_3_* and *TlNbF_3_* in both (spin-up plus spin-down) alignments.

**Figure 5 materials-15-05684-f005:**
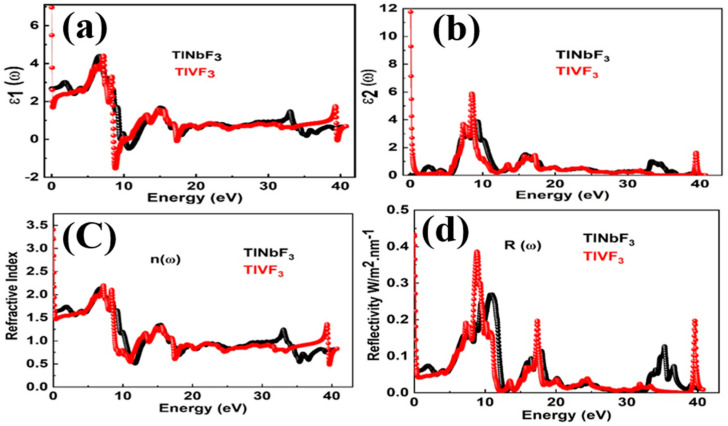
The calculated $\varepsilon_{1}\left(\omega \right)$, $\varepsilon_{2}\left(\omega \right)$, n(ω) and R(ω) of *TlVF_3_* and *TlNbF_3_*.

**Figure 6 materials-15-05684-f006:**
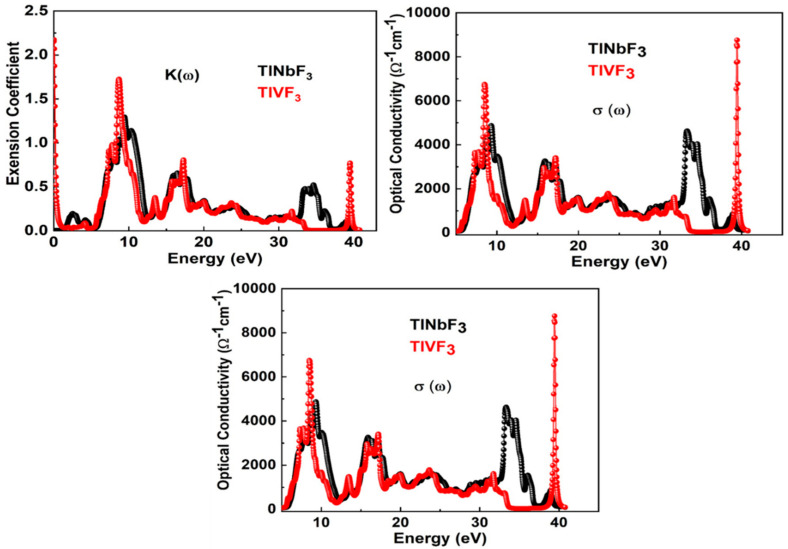
The calculated K(ω), and σ(ω) of $TlVF_{3}$ and $TlNbF_{3}$ compounds.

**Table 1 materials-15-05684-t001:** Structural parameters of *TlVF*_3_ and *TlNbF*_3_ using TB-mBJ approach in which $a_{0}$ shows optimized lattice constant in angstrom $\left(Å\right)$, B is the bulk modulus in GPa, B′ depict the derivative of bulk modulus in GPa, and V_o_ in (a.u)^3^ is the optimized unit cell volume.

Compounds	a_o_ (Å)	B (GPa)	B′ (GPa)	V_o_ (a.u)^3^
** *TlVF_3_* **	**4.638**	**63.429**	**4.272**	**513.37**
** *TlNbF_3_* **	**4.372**	**109.561**	**5.544**	**616.26**

**Table 2 materials-15-05684-t002:** The calculated elastic parameters *C*_11_, *C*_12_, *C*_44_, B, A, G, E (all in GPa), υ, and B/G of both the $TlVF_{3}$ and $TlNbF_{3}$ compounds. The table shows the cubic elastic constant *C*_11_, *C*_12_, *C*_44_ and the bulk modulus “B” shear modulus “G”, Young’s modulus in GPa, the anisotropy factor “A”, the Poisson ratio “υ”, and the Pugh ratio “B/G”.

Compounds	$C_{11}$	$C_{12}$	$C_{44}$	B	A	G	E	υ	B/G
** *TlVF_3_* **	**131.3907**	**27.3113**	**−3.8737**	**63.429**	**−0.0744**	**5.8491**	**17.0243**	**0.6725**	**10.8441**
** *TlNbF_3_* **	**168.0201**	**77.5591**	**−17.2098**	**109.561**	**−0.3804**	**−15.332**	**−48.2495**	**0.8816**	**7.1454**

**Table 3 materials-15-05684-t003:** Investigated magnetic moments for *TlNbF_3_* and *TlVF*_3_ compounds.

Site	*TlNbF_3_*	*TlVF_3_*
**Tl**	0.076	0.083
**F**	0.087	0.068
**V**	0	2.47
**Nb**	3.063	0
**Interstitial site**	0.894	0.673
**Total**	4.12	3.294

## Data Availability

All the obtained data is provided in the research article.

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
