# Peer review of "Insight into the Structural, Electronic, Elastic, Optical, and Magnetic Properties of Cubic Fluoroperovskites ABF3 (A = Tl, B = Nb, V) Compounds: Probed by DFT"

_materials, 2022, doi:10.3390/ma15165684_

Round 1

Reviewer 1 Report

The manuscript presents a conventional DFT study of two compounds TlVF3 and TlNbF3. The authors have investigated a series of properties including crystal structures, elastic constants, electronic band structure, as well as spin moments. However, it seems to me that the paper was prepared casually and not suitable for publication until the following issues have been resolved. 

(1). On Page 1, in the abstract, authors should avoid using "the first theoretical study", because if this study is not novel, it is not necessary to get published.

(2). On Page 2, the "Introduction" section appears pretty poor. I think an effective introduction needs to get the reader's attention in a logical order:

(a) Why are the so-called fluoro-perovskites ABF3 compounds interesting?

(b) What has been done in this field, what is known, what is unknown, how

       does the present work contribute to this field?

(c) Why do the authors have to use the TB-mBJ method instead of the other more popular LDA or GGA exchange-correlation functionals?

(3). The authors have chosen not to show the thermodynamic properties such as the cohesive energy. It would be difficult to characterize the thermal stability of these two compounds especially when experimental study is not available either.

(4). On Page 3, in Table 1, what does a0 stand for, and what do B and B' refer to?  Some units are given, while others are missing. The physical meaning of these parameters should be stated clearly in the caption. 

(5). On Page 6, in Table 3, what is the 'Interstitial site'? It is a bit surprising to see such site also carry a sizable local spin moment.

(6). On Page 6, lines 166-168, can authors explain why total magnetic moment greater than 1 indicate strong ferromagnetic exchange interaction? 

(7). In addition, there are lots of citation warning errors of figures in the manuscript, which need to be cleared in future submissions.

Author Response

Response to Reviewers

            We thank the reviewers for their very nice comments on our manuscript. Following their kind suggestions, we have made modifications to our manuscript and highlighted them. Hope this could improve the readability. The response to each comment is highlighted as blue and highlighted yellow in the manuscript.

Response to Reviewer 1:

[1] We thank the reviewer for such a nice comment. “On Page 1, in the abstract, authors should avoid using "the first theoretical study", because if this study is not novel, it is not necessary to get published.

Following the very nice suggestion of the reviewer, we avoid using "the first theoretical study" in the abstract and wherever is needed in the manuscript.

[2] We again thank the reviewer for such a nice comment. “On Page 2, the "Introduction" section appears pretty poor. I think an effective introduction needs to get the reader's attention in a logical order”.

(a) Why are the so-called fluoroperovskites ABF3 compounds interesting?

(b) What has been done in this field, what is known, what is unknown, and how does the present work contribute to this field?

(c) Why do the authors have to use the TB-mBJ method instead of the other more popular LDA or GGA exchange-correlation functionals?

Following the very nice suggestion of the reviewer, we have modified the introduction section of the manuscript and added the answer to the points (a), (b), and (c) raised by the respected reviewer. Hopefully, now the readers get an effective introduction.

Recently the most prevalent and extensively researched structure in materials science is the perovskite structure. Fluoroperovskites ABF3 compounds are interesting nowadays for researchers due to their wide applications in many semiconducting industries, solar cells industries, and other electronic gadgets. The fluoroperovskites with A and B are metallic cations and F which is a highly electronegative anion possesses various structural, electronic, elastic, thermoelectric, thermodynamics, and magnetic properties because of its variant electronic bands gaps. The perovskite compounds have the general chemical formula ABX3, where A and B are two cations of various sizes, and X is an anion linked to both of them. Its ideal structure is cubic, and the B atoms in a typical anionic octahedron are located in the middle. This atomic configuration may appear simple, but it conceals a variety of unique physical and chemical features. The selected fluoroperovskite  (A = Tl, B = Nb, V) compounds possess acubic symmetry. Lee et al., [1] and Manivannan et al., [2] confirmed this experimentally. In the space group, “A” and “B” atoms (cations) occupy the edges and body center positions respectively while the face centers are occupied by “F” atoms (anion) [3]. It is reported that most of the cubic fluoroperovskites compounds are elastically anisotropic and mechanically stable and also possess interesting electronic properties and magnetic properties [4]. Due to their attractive properties, these materials have a variety of applications, including photovoltaic, vehicle energy, device, and in lenses industry as well as their transparent characteristics are used in antireflection coatings [5-9]. Woodward and Lufaso reported that cubic perovskite can transfer to other forms of crystal structures [10]. The mixture of F with either organic or inorganic and transition metals form stable fluoroperovskites [11]. In this work, the structural, elastic, electronic, and optical properties within the spin-polarized case of  and compounds have been studied theoretically by DFT (TB-mBJ method) using the WIEN2K computational simulation code. The method of TB-mBJ potential is used because of its accuracy in the electronic band gaps. As the LDA or GGA exchange-correlation, functional underestimate the electronic band gaps. These compounds have not been studied theoretically or experimentally before. Therefore, this work can be used as a reference for further studies of such types of compounds.               

[3] We again thank the reviewer for such a nice comment. “The authors have chosen not to show the thermodynamic properties such as the cohesive energy. It would be difficult to characterize the thermal stability of these two compounds especially when the experimental study is not available either”.

This is a very nice suggestion from the reviewer. In this study, our focus is on the computation of structural, electronic, elastic, optical, and magnetic properties in spin-polarized configurations for cubic fluoroperovskites  (A = Tl, B = Nb, V) compounds. As we predict that the interesting compounds are structurally stable. We intend to work on the thermodynamics properties of these selected materials in the future.

[4] We again thank the reviewer for such a nice comment. “On Page 3, in Table 1, what does a0 stand for, and what do B and B' refer to?  Some units are given, while others are missing. The physical meaning of these parameters should be stated clearly in the caption. 

Following the very nice suggestion of the reviewer, we have mentioned the units and meaning of each parameter in Table 1, on page 3.

Table 1. Structural parameters of TlVF3 and TlNbF3 using TB-mBJ approach in which, shows optimized lattice constant in angstrom, B is the bulk modulus in GPa, B/ depict the derivative of bulk modulus in GPa, and Vo in (a.u)3 is the optimized unit cell volume.

Compounds

ao ()

B (GPa)

B/ (GPa)

Vo (a.u)3

TlVF3

4.638

63.429

4.272

513.37

TlNbF3

4.372

109.561

5.544

616.26

[5] We again thank the reviewer for such a nice comment. “On Page 6, in Table 3, what is the 'Interstitial site'? It is a bit surprising to see such site also carry a sizable local spin moment”.

This is a very nice suggestion of the reviewer, the interstitial site is the set of points of space that are not in any of the atomic spheres. So, the spin interstitial magnetic moment is the difference between the number of spin-up and spin-down electrons in the interstitial. Such sites also carry sizable local spin magnetic moments in different ranges.

[6] We again thank the reviewer for such a nice comment. “On Page 6, lines 166-168, can the authors explain why a total magnetic moment greater than 1 indicates strong ferromagnetic exchange interaction?

Following the very nice suggestion of the reviewer, on page 6, the threshold value for the indication of ferromagnetic materials is "1". Materials that possess a greater total magnetic moment than the threshold value will indicate strong ferromagnetism. As in our case, both the materials have magnetic moment greater than “1” and thus are strong ferromagnetic exchange interactions.

[7] We again thank the reviewer for such a nice comment. “In addition, there are lots of citation warning errors of figures in the manuscript, which need to be cleared in future submissions”.

Following the very nice suggestion of the reviewer, we have cleared the citation warning errors of figures in the manuscript.

Reviewer 2 Report

Insight into the structural, electronic, elastic, optical and magnetic properties of cubic fluoro-perovskite ???? (A = Tl, B = 3 Nb, V) compounds: Probed by DFT. The paper and the topic of ABF3 perovskite solar cell are quite interesting but the results need to be improved. There are some minor miss understanding that require some modifications in the paper. After the following corrections paper may be acceptable for publication:

1. The English and grammatically mistakes should be revise carefully and the monoculture need improvement.

2.The abstract should contain the results obtained in the manuscript.
3. the Introduction part needs to be developed and The ideal fluoro-perovskites with the general formula
??F3, the authors can add the A, B cations exemple :

Suggested references:

https://doi.org/10.1063/1.5078907,      

https://doi.org/10.1080/00150199008223849,

https://doi.org/10.3389/fenrg.2022.840817,

4 . the originality of the work should be specified clearly and correct this sentences:  unit cells for both the cubic fluoro-perovskites ????3 and ?????3 having space 79 group Pm-3m (#221) are shown in the Error! Reference source not found and [ It can be observed in the Error! Reference 122 source not found. that the obtained values of the anisotropy factors are -0.0744 and -0.3804}

5 .the manuscript is full of some errors.

6.  Structural parameters of TlVF3 and TlNbF3 using TB-mBJ approach on the table 1 should be explain based on both structure  of Perovskite.

Author Response

Response to Reviewers

            We thank the reviewers for their very nice comments on our manuscript. Following their kind suggestions, we have made modifications to our manuscript and highlighted them. Hope this could improve the readability. The response to each comment is highlighted as blue and highlighted yellow in the manuscript.

Response to Reviewer 2:

[1] We thank the reviewer for such a nice comment. “The English and grammatical mistakes should be revised carefully and the monoculture needs improvement”.

Following the very nice suggestion of the reviewer, we have carefully revised the whole manuscript and removed the English and grammatical mistakes wherever needed. Now hopefully the readability of the manuscript will bear well.

[2] We again thank the reviewer for such a nice comment. “The abstract should contain the results obtained in the manuscript”.

Following the very kind suggestion of the reviewer, our manuscript contains the obtained results as described in the abstract:

This work presents the structural, electronic, elastic, optical, and magnetic properties in spin-polarized configurations for cubic fluoroperovskite  (A = Tl, B = Nb, V) compounds studied by density functional theory (DFT) using Tran-Blaha-modified Becke-Johnson (TB-mBJ) approach. The ground state characteristics of these compounds i-e the lattice parameter, bulk modulus (B), and its pressure derivative  are investigated. The structural properties depict that these compounds possess a cubic crystal structure and have stable ground state energy. Electronic-band structures and density of states (DOS) in both spin-polarized cases are studied which reports the semiconducting nature of both materials. The total density of states (TDOS) and partial density of states (PDOS) studies in both spin configurations show that the maximum contribution of states to the various bands is due to the B-site (p-states) atoms as well as F (p-states) atoms. Elastic properties including anisotropy factor (A), elastic constants i-e C11, C12, and C44, Poisson's ratio (Ê‹), shear modulus and (G), Young's modulus (E) are calculated. In terms of elastic properties, the higher "B" (bulk modulus) and B/G ratio yield that these compounds exhibit a ductile character. Magnetic properties indicate that both the compounds are ferromagnetic. In addition, investigations of the optical spectra such as the real () and imaginary () parts of the dielectric function, refractive index, optical reflectivity , optical conductivity , absorption coefficient , energy loss function  and electron extinction coefficient  are carried out which shows the transparent nature of  and. Based on the reported research work on these selected materials, their applications can be predicted in many modern electronic gadgets.

[3] We again thank the reviewer for such a nice comment. “The introduction part needs to be developed and the ideal fluoroperovskites with the general formula ??F3, the authors can add the A, B cations example:

Suggested references:

https://doi.org/10.1063/1.5078907,

https://doi.org/10.1080/00150199008223849,

https://doi.org/10.3389/fenrg.2022.840817

Following the very kind suggestion of the reviewer, we have developed an introduction part of the manuscript.

Recently the most prevalent and extensively researched structure in materials science is the perovskite structure. Fluoroperovskites ABF3 compounds are interesting nowadays for researchers due to their wide applications in many semiconducting industries, solar cells industries, and other electronic gadgets. The fluoroperovskites with A and B are metallic cations and F which is a highly electronegative anion possesses various structural, electronic, elastic, thermoelectric, thermodynamics, and magnetic properties because of its variant electronic bands gaps. The perovskite compounds have the general chemical formula ABX3, where A and B are two cations of various sizes, and X is an anion linked to both of them. Its ideal structure is cubic, and the B atoms in a typical anionic octahedron are located in the middle. This atomic configuration may appear simple, but it conceals a variety of unique physical and chemical features. The selected fluoroperovskite  (A = Tl, B = Nb, V) compounds possess acubic symmetry In the space group, “A” and “B” atoms (cations) occupy the edges and body center positions respectively while the face centers are occupied by “F” atoms (anion) [1]-[3]. It is reported that most of the cubic fluoroperovskites compounds are elastically anisotropic and mechanically stable and also possess interesting electronic properties and magnetic properties [4]. Due to their attractive properties, these materials have a variety of applications, including photovoltaic, vehicle energy, device, and in lenses industry as well as their transparent characteristics are used in antireflection coatings [5-9]. Woodward and Lufaso reported that cubic perovskite can transfer to other forms of crystal structures [10]. The mixture of F with either organic or inorganic and transition metals form stable fluoroperovskites [11]. In this work, the structural, elastic, electronic, and optical properties within the spin-polarized case of  and compounds have been studied theoretically by DFT (TB-mBJ method) using the WIEN2K computational simulation code. The method of TB-mBJ potential is used because of its accuracy in the electronic band gaps. As the LDA or GGA exchange-correlation, functional underestimate the electronic band gaps. These compounds have not been studied theoretically or experimentally before. Therefore, this work can be used as a reference for further studies of such types of compounds.

We have also included in the manuscript the suggested references by the reviewer:

  1. D. A. A. Ahmed, S. BaÄŸcı, E. Karaca, and H. M. Tütüncü, "Elastic properties of ABF3 (A: Ag, K, and B: Mg, Zn) perovskites," in AIP Conference Proceedings, 2018, vol. 2042, no. 1, p. 20035.
  2. L. L. Boyer and P. J. Edwardson, “Perovskite to antiperovskite in abf3 compounds,” Ferroelectrics, vol. 104, no. 1, pp. 417–422, 1990.
  3. A. Bouch, J. Marí-Guaita, B. Sahraoui, P. Palacios, and B. Marí, "Tetrabutylammonium (TBA)-Doped Methylammonium Lead Iodide: High Quality and Stable Perovskite Thin Films," Front. Energy Res, vol. 10, p. 840817, 2022.

[4] We again thank the reviewer for such a nice comment. “The originality of the work should be specified clearly and correct in these sentences:  unit cells for both the cubic fluoroperovskites ????3 and ?????3 having space 79 group Pm-3m (#221) are shown in the Error! Reference source not found and [It can be observed in the Error! Reference 122 sources not found. that the obtained values of the anisotropy factors are -0.0744 and -0.3804}

Following the very kind suggestion of the reviewer, we have specified the originality of our work and fixed the Error! Reference source not found in the manuscript.

[5] We again thank the reviewer for such a nice comment. “The manuscript is full of some errors”.

Upon the very kind suggestion of the reviewer, we have removed the errors in the manuscript wherever needed.

[6] We again thank the reviewer for such a nice comment. “Structural parameters of TlVF3 and TlNbF3 using TB-mBJ approach on table 1 should be explained based on both structure of Perovskite”

Following the very kind suggestion of the reviewer, we have explained the structural parameters in table 1 based on the structure of both perovskite compounds.

Figure 1. Crystal structures of  (A = Tl, B = Nb, V) compounds.

Round 2

Reviewer 1 Report

Most of my concerns have been addressed. I recommend the revised manuscript for publication.